# Determinants of Undernutrition and Associated Factors of Low Muscle Mass and High Fat Mass among Older Men and Women in the Colombo District of Sri Lanka

**DOI:** 10.3390/geriatrics7020026

**Published:** 2022-02-28

**Authors:** Samantha Chandrika Vijewardane, Aindralal Balasuriya, Phyo Kyaw Myint, Alexandra M. Johnstone

**Affiliations:** 1Nutrition Division, Ministry of Health, Colombo 00600, Sri Lanka; 2Ageing Clinical & Experimental Research Team, Institute of Applied Health Sciences, School of Medicine, Medical Sciences and Nutrition, University of Aberdeen, Aberdeen AB25 2ZD, UK; phyo.myint@abdn.ac.uk; 3Department of Public Health and Family Medicine, Faculty of Medicine, General Sir John Kotelawala Defense University, Dehiwala-Mount Lavinia 10390, Sri Lanka; dr.balasuriya@yahoo.com; 4The Rowett Institute, School of Medicine, Medical Sciences and Nutrition, University of Aberdeen, Aberdeen AB25 2ZD, UK; alex.johnstone@abdn.ac.uk

**Keywords:** ageing adults, body composition, fat mass, skeletal muscle mass, undernutrition

## Abstract

Undernutrition is a health challenge due to an expanding older population. The aims of the study were to assess the prevalence and determinants of undernutrition and, associated factors of low muscle and high fat mass among older men and women in the Colombo district of Sri Lanka. A cross sectional study was conducted using a multistage cluster sampling technique. Undernutrition was defined based on anthropometry and body composition assessed using bio-electrical impedance. Sex-specific multivariable logistic regression analyses were conducted. Of 800 participants (30.6% men), 35.3% were undernourished. The factors significantly associated with undernutrition among older women were hypertension with an adjusted odds ratio (aOR) (1.97; 1.36–2.88) and musculoskeletal disabilities aOR (2.19; 1.36–3.53). Among women, age ≥ 70 (1.79; 1.18–3.34) and diabetes (1.77; 1.10–2.84) were associated with low muscle mass and age ≥ 70 (2.05; 1.21–3.47), diabetes (2.20; 1.35–3.59) and disability in chewing (2.39; 1.30–4.40) were associated with high fat mass. Among men, age ≥ 70 years, no/up to grade 5 education, diabetes, visual disability, little/no responsibility in food shopping and not getting nutritional advice from media were associated with reduced odds of low muscle mass and no/up to grade 5 school education, disability in chewing and little/no responsibility in food shopping were associated with reduced odds of high fat mass. Undernutrition among older people is common in Sri Lanka. We have identified key factors associated with low muscle mass and high fat mass in this setting. Given the potential consequences of these conditions, our study provides potential targets for prevention of undernutrition and sarcopenic obesity.

## 1. Introduction

The World Health Organization (WHO) defines undernutrition as a lack of proper nutrition, resulting from not having enough food, and not eating adequate nutritious food, due to physical or psychological causes [1]. With the global demographic transition towards ageing populations, undernutrition in older age has become a global challenge as a major contributor to morbidity and mortality. Consequently, the health and economic burden associated with undernutrition imposes a public health challenge and causes concerns, especially in low- and middle-income countries, such as Sri Lanka.

Undernutrition is a major contributing factor for reduced immunity and increased susceptibility to infections, increased number of hospital admissions, poor recovery from illness and longer hospital admission. As older people are a group with nutritional vulnerability, they are more prone to be undernourished [2]. Sarcopenia is a condition characterized by low skeletal muscle quantity, quality and thus strength [3]. This is linked to difficulty in carrying out activities of daily living and leading to a higher risk of disability, frailty and falls [4]. Another important aspect is that sarcopenia is characterized by an increase in fat mass with a reduction in muscle mass [3].

Obesity is a chronic disease and another form of malnutrition, characterized by high body fat mass and increased risk of development of metabolic diseases, cardiovascular diseases and some cancers [5]. The WHO recommends using the criteria of BMI > 30 kg/m^2^ to define obesity. Although BMI is a useful population level indicator to diagnose and track global trends in obesity, the measurement of body fat mass has a better predictive ability than BMI for the assessment of individual risk of metabolic syndrome and cardiovascular diseases [5].

Sarcopenic obesity is an emerging nutritional problem among older people [3]. It is linked to age-related changes in biological pathways including lowering of metabolic rate, hormone status and thermogenesis, all of which can impact on body composition by lowering muscle mass and increasing fat mass. Due to the trends toward an older population with obesity pandemic, there has been a rise in this phenotype [5]. It is presented as a combination of sarcopenia and over/malnutrition (obesity).

When the intake of energy and protein is not adequate to meet individual needs, body fat and muscle are catabolized to provide energy with consequent symptoms, such as fatigue or lack of energy [6]. Undernutrition is a condition driven by multiple factors including socio-demographics, physiological, pathological, financial and lifestyle factors [7]. Physiological changes which accompany an advancing age, such as dental status, impaired vision, taste or smell and poor appetite in older age, also contribute to insufficient dietary and nutritional intake [8].

The prevalence of undernutrition in Sri Lanka has previously been reported as being highly prevalent across the population, measured by using anthropometry or the Mini Nutritional Assessment tool (MNA) [9]. The estimated prevalence of undernutrition was 21–67% in a hospital setting [10] and 30% in elderly homes [8]. Within the community dwelling setting, wide variation can be observed in the prevalence of undernutrition in older people living in Sri Lanka and reported to be as high as 45% in some regions [11,12,13,14]. Most studies have applied anthropometric measurements alone or in combination with the MNA tool to assess undernutrition [10,11]. We have applied the new gold standard criteria described by the international nutritional societies, ASPEN or ESPEN, in assessing undernutrition [15] and explored the determinants of undernutrition in older Sri Lankans.

At the Global Leadership Initiative on Malnutrition (GLIM), all the major nutrition societies agreed on a phenotypic criterion, which includes low BMI and low muscle mass in diagnosing malnutrition [15]. BMI and fat mass were used (alone or in combination) in diagnosing malnutrition both in the clinical care setting and the community setting [11,13]. According to the WHO, mid upper arm circumference (MUAC) is an important indicator for estimating lean body mass linked to malnutrition [15]. Measuring BMI among older people has limitations since conditions such as kyphosis in old age can cause loss of height, loss of muscle tone and change in posture can also contribute to reduced height and presence of oedema/ascites can give rise higher body weight and therefore skewed BMI values [16]. BMI may not purely reflect adiposity as body weight is used as the numerator which is composed of both fat and fat-free mass [17]. The MNA tool does not consider body composition assessment, which was strongly recommend by the ASPEN and ESPEN criteria [14]. The study team of experts came to a consensus to apply the composite criteria of low BMI, low MUAC, low muscle and fat mass as criteria in the present study (Appendix A). As shown in Appendix A, most previous studies conducted either locally or internationally have used anthropometry or MNA alone to assess undernutrition. We therefore aim to fill this evidence gap by using a combination of anthropometry and body composition to assess undernutrition. Since the body composition changes with increasing age, low skeletal muscle mass and high fat mass is evident among older people and the sex and ethnicity are important determinants of this change [17,18].

The primary aim of this study was to assess the prevalence and determinants of undernutrition and the secondary aim was to further explore the sex-specific factors associated with low skeletal muscle mass and high fat mass among older men and women in the Colombo district of Sri Lanka.

## 2. Materials and Methods

A cross-sectional analytical study was carried out in Colombo District, Sri Lanka. Colombo District has both urban and rural populations with diverse socio-economic composition and it has 13 administrative units called Divisional secretariat (DS) areas, and each DS area consists of administrative sub-units called Grama Niladari Divisions (GN) [19].

Ethical approval was obtained from the Ethical Review Committee, Faculty of Medicine, University of Kelaniya, Sri Lanka (P 123/6/2018).

### 2.1. Study Population

Older people aged ≥ 60 years, who were residing in the area for the last 3 months at the time of the study (March 2019) were invited to participate.

Those who were institutionalized or attending day care centers, suffering from cancer, neurodegenerative disorders or chronic renal failure, who were on pacemakers and deemed unable to provide informed consent (e.g., severe cognitive deficit), were excluded from the study.

Eligible subjects were provided with study information both verbally and in written format and gave informed consent.

### 2.2. Sample Size Calculation

The sample size was calculated by considering the estimated prevalence of undernutrition as 20%, desired level of precision as 5%, with 95% as the desired level of confidence and design effect as 2.9. After adding a non-response rate of 10%, final total sample size was 800.

### 2.3. Sampling Technique

A multi-stage cluster sampling technique was used. Out of 13 DS Divisions in the Colombo District, 7 were selected randomly. The GN divisions within these selected 7 DS divisions with their population of older people aged ≥ 60 years were listed according to alphabetical order. The technique of probability proportionate to the size was utilized in selecting 40 clusters from 7 DS divisions by applying the following sampling interval. One cluster was equal to one GN area and the cluster size was fixed at 20. The first house to be visited from the cluster was chosen from the voters’ register using a random number table. The subsequent households to be visited were defined as to the nearest to the right side of the preceding house. The data collectors visited the first house and inquired whether an eligible person resided in the household. If that person was willing to participate in the study, he/she was included and necessary instructions were provided and revisited after one week to collect anthropometric and body composition measurement data. The data collectors visited households using the aforementioned method until they had recruited 20 older people in each cluster. If there were more than one older person in one household, one person was selected randomly.

### 2.4. Data Collecting Instruments

An interviewer administered the questionnaire (Appendix A), anthropometric measurements and body composition measurements assessed using bio-electrical impedance (BIA) were used to collect data.

### 2.5. Assessment of the Nutritional Status

Anthropometry included body weight (kg) and standing height (cm) to calculate body mass index (BMI), mid upper arm circumference (MUAC) and BIA measurements using a compartment model of body composition (skeletal muscle mass and body fat mass).

A standard operating procedure was applied for weight and height measurements by trained assistants. Data collection was performed in participants’ own homes. Standing height was measured using a Harpenden pocket Stadiometer (Chasmors Ltd., London, UK) to the nearest 0.1 cm. Weight was recorded using calibrated Omron scales (HBF-212, Kyoto, Japan) to the nearest 100 g. BMI was calculated from body mass and height. The MUAC is measured at its mid-point between the tip of the acromion and olecranon process in the non-dominant hand. It was measured using a non-stretchable tape to the nearest 0.1 cm.

Body composition was estimated using commercially available single frequency, four electrode bio-impedance analyzer (HBF-212, Japan). The participants were instructed to drink an adequate amount of water and consume a normal diet during the preceding week, not to have any vigorous physical exercise during the preceding 12 h and avoid alcohol during the preceding 48 h to maintain normal hydration of the body. They were asked to not eat/drink during the preceding 4 h, empty their bladders just before the measurement and those with pacemakers were excluded from the study. All metal objects were removed prior to measurements and all the procedures were carried out according to the manufacturer’s manual. The participant was asked to stand straight on the BIA machine with the feet touching the electrodes, with hands on either side of the body. During the measurement, the analyzer recorded the whole-body impedance from foot to foot by applying an electric alternating current flux of 0.8 mA at an operating frequency of 50 kHz. The percentage of muscle mass and fat mass were calculated from the whole-body impedance value and pre-entered personal data (age, gender, height, weight) using a standard equation provided by the manufacturer. Inter-operator reliability and precision of measurements was assured by regular monitoring and supervision.

BIA has been previously validated for application in older populations [20] and the machine used in our study (Omron, HBF-212, Japan) has been applied for 2 compartment body composition assessments in older populations [21]. Although BIA is the most commonly used method to assess the body composition in community studies, the equations chosen to transform impedance values may not be entirely valid as the equation is not validated for application in a Sri Lankan population [20].

We assessed the presence of diabetes, hypertension, COPD and disabilities by asking the participant/caregiver which was verified from clinic/hospital records. A working definition was used to define musculoskeletal disabilities. “Injuries/disorders of muscle/bone with limitation of movements”. Operationalized definition of variables is included in Appendix A.

### 2.6. Criteria of Classification of the Older People as Undernourished

Participants were categorized as undernourished according to the BMI and MUAC classifications by the WHO in 2000 [22] and 1995 [23], respectively, and for percentage of body fat the criteria described by Gallagher et al., for the older population in Asia [24] was used, whilst muscle mass was defined using values obtained by Tichet et. al. [25] (Appendix A).

The questionnaire was developed to cover potential confounders of undernutrition in a Sri Lankan setting using a consensus approach. The variables included in the study were classified as socio-demographic, socio-economic, co-morbidities, physiological disabilities, diet and lifestyle.

### 2.7. Data Collection

Data collection was conducted using trained data collectors under the supervision of the principal investigator (PI). Two medical doctors were recruited for this purpose. They were trained by the PI on recruiting the participants to the study, taking anthropometric and body composition measurements using standard operating procedure. Two sets of properly calibrated measuring equipment were used to assure the minimum variation of the measurements. Each measurement was made twice and the average value of the two measurements was considered as the final value. Inter-observer reliability of anthropometric measurements against PI was assessed by remeasuring height and weight in 5% (*n* = 40) of participants in the sample by the PI. Intra-observer reliability was assessed by re-measuring the height and weight in 5% (*n* = 40) of participants in the sample by the same data collector on two occasions. Significant levels of agreement were observed for all instances. The Pearson correlation co-efficient (r) and *p* values were r = 0.88 (*p* = 0.001), r = 0.83 (*p* = 0.001), r = 0.91 (*p* = 0.001) and r = 0.81 (*p* = 0.001) for measures of inter-observer and intra-observer reliability for height and weight, respectively.

### 2.8. Data Analysis

The data analysis was carried out using SPSS (version 22.0). Normal distribution of the variables was checked visually by histograms. The determinants of undernutrition and factors associated with low muscle mass and high fat mass were analyzed using bivariate cross tabulations and expressed as odds ratios (unadjusted) and 95% confidence intervals. All the variables were included in the multivariable logistic regression analysis to minimize the confounding factors. Sex specific analyses were conducted to characterize those with low skeletal muscle mass and high fat mass. Among older men and women, a cut-off point for low skeletal muscle mass (sarcopenia) was taken as ≤25th centile, and high fat mass (obesity) was taken as ≥75th centile. Those who were classed with both low skeletal mass and high fat mass were considered to have sarcopenic obesity. Effect size was reported using Phi coefficient in multiple comparison of factors reported using chi-square, and Cohen’s d in univariate analysis reported using ORs.

## 3. Results

A total of 800 older people participated in the study. The mean age of the sample was 68.1 (SD = 5.8, range 60–94) years, with more women 69.4% (*n* = 555). In the sample, mean BMI was 24.0 kg/m^2^ (SD = 4.21), mean MUAC was 28.86 cm (SD = 3.12), mean muscle mass was 26.08% (SD = 3.64) and mean fat mass was 32.54% (SD = 7.01).

### 3.1. Multiple Comparisons of Factors among Older Men and Women in the Sample

As anticipated, there were significant differences in many factors between the men and women (Table 1).

### 3.2. Prevalence of Undernutrition among the Older People

We report that 35.3% (*n* = 282; 95% CI: 31.8–38.7%) were undernourished using the composite criterion (Appendix A). More women were classified as undernourished (47.2%, *n* = 262), in contrast to the older men (8.2%, *n* = 20) using this criterion. Interestingly, when body composition assessment was applied for this group, 25.7% (*n* = 143) of the older women and 25.7% (*n* = 63) of older men were classified as having sarcopenia, 25% (*n* = 139) of the older women and 26.9% (*n* = 66) of older men were classified as having obesity and 4.3% (*n* = 24) of the older women and 15.1% (*n* = 37) of older men were classified as having sarcopenic obesity (Appendix A).

### 3.3. Determinants of Undernutrition among Older Men and Women

Table 2 shows that after adjusting for confounders, the factors significantly associated with undernutrition among the older women were, having hypertension (aOR = 1.97; 95% CI = 1.36–2.88) and musculoskeletal disabilities (aOR = 2.19; 95% CI = 1.36–3.53). Low numbers of male participants yielded insufficient data to run the regression analysis (Table 2).

### 3.4. Factors Associated with Low Skeletal Muscle Mass among Older Men and Women

Table 3 displays data after adjusting for confounders. Among older women, age ≥ 70 years (aOR = 1.99; 95% CI = 1.18–3.34) and having diabetes mellitus (aOR = 1.77; 95% CI = 1.10–2.84) were significantly associated with low skeletal muscle mass. For the men, ≥70 years (aOR = 0.43; 95% CI = 0.20–9.1), no education or up to grade 5 (aOR = 0.27; 95% CI = 0.09–0.76), presence of diabetes (aOR = 0.34; 95% CI = 0.15–0.77), disability in vision (aOR = 0.24; 95% CI = 0.08–0.69), little/no responsibility in food shopping (aOR = 0.11; 95% CI = 0.01–0.67) and not getting nutritional advice from media (aOR = 0.07; 95% CI = 0.01–0.95) were significantly associated with reduced odds of low muscle mass.

### 3.5. Factors Associated with High Fat Mass among Older Men and Women

Table 4 reports that after adjusting for confounders, among older women, age ≥ 70 years (aOR = 2.05; 95% CI = 1.21–3.47), having diabetes mellitus (aOR = 2.20; 95% CI = 1.35–3.59) and disability in chewing (aOR = 2.39; 95% CI = 1.30–4.40) were significantly associated with high fat mass. In males, no education or up to grade 5 (aOR = 0.29; 95% CI = 0.10–0.85), disability in chewing (aOR = 0.34; 95% CI = 0.13–0.94) and little or no responsibility in food shopping (aOR = 0.05; 95% CI = 0.01–0.35) were significantly associated with reduced odds of high fat mass.

## 4. Discussion

We have identified several factors that are linked to undernutrition among community dwelling older adults in the Sri Lankan population, in Colombo district with diverse demographic background. Figure 1 shows the summary of the factors associated with undernutrition, low muscle and high fat mass among older men and women. Hypertension and musculoskeletal disorders, such as arthritis and muscular dystrophies, were significantly associated with undernutrition among community living older women in Sri Lanka. Low skeletal muscle mass and high fat mass was evident among the older women ≥ 70 years and those with diabetes mellitus. Disability in chewing was also significantly associated with high fat mass among older women.

The current study identified a number of key factors associated with having a significant inverse association with low muscle mass in men, which included, age ≥ 70 years, no school education or up to grade 5, presence of diabetes, disability in vision, little or no responsibility in food shopping and not getting nutritional advice from media. No school education or up to grade 5, disability in chewing and little or no responsibility in food shopping were identified as having significant inverse association with high fat mass in men. These factors have important implications for targeting public health policies in Sri Lanka.

There is rising public health concern over the effects of rapidly growing older populations. Undernutrition and sarcopenia are both recognized as important risk factors for poor health outcomes [1,4]. By identifying determinants of undernutrition, our study informs potential public health policies.

### 4.1. Prevalence of Undernutrition in Sri Lanka

We found that the prevalence of undernutrition among the older people as 35.3% using composite criterion. In contrast, other studies which used BMI alone showed varying estimates ranging from 20% to 39% [11,13]. Prevalence of undernutrition using the MNA tool, which assesses undernutrition based on anthropometry, recent weight loss, reduced mobility, decrease in food intake and acute disease or psychological stress over the past three months, also appears to underestimate the prevalence in Asia relative to the current study. The reported prevalence was around 19.5% in India [26] and 24.0% in Nepal [27]. These data compared to much lower prevalence data observed in more developed countries such as France (7.5%) and Turkey (16%) highlight the need for urgent actions to address the impact of undernutrition in low-middle income countries by addressing its potential risk factors [28,29].

### 4.2. Identification of Factors Associated with Undernutrition

Of the socio-demographic factors, age ≥ 70 years was significantly associated with low skeletal muscle mass and high fat mass among the older women in the present study. Age consistent findings for these associations were reported in descriptive studies conducted among the older people of ≥60 years reported in Galle [30] and Kandy district [12] where advancing age was significantly associated with undernutrition.

Throughout the life course, women tend to carry more adiposity than males and their muscle percentage is relatively lower [18]. The majority of women (70.6%) in the current study had high muscle mass and high fat mass. Among the male population, majority (62.4%) had high muscle mass and low-fat mass. However, sarcopenic obesity is more prevalent among men (15.1%) than in women (4.3%). In a study conducted in Korea, 0.8% of older women and 1.3% older men had sarcopenic obesity among samples of 328 and 198 older people, respectively [31]. This pattern of gender difference in prevalence has been reported elsewhere in other regions of the world [32,33,34]. The proposed mechanism suggests that whilst older men may have more muscle mass than older women, muscle deterioration is faster in men. This has an important impact on healthy ageing.

Regarding co-morbid conditions, the presence of diabetes mellitus was a significantly associated factor for low skeletal muscle mass and high fat mass among older women, consistent with the findings of another study conducted in Sri Lanka [12]. It is evident that there is a greater gain of fat mass and loss of skeletal muscle mass among older people with diabetes compared to those without diabetes [18]. In the sample, 29.4% (*n* = 42) and 39.69% (*n* = 163) of older women with low muscle mass and high muscle mass (*p* = 0.03) and 27.3% (*n* = 38) and 40.1% (*n* = 167) of older women with high fat mass and low-fat mass (*p* = 0.01) had diabetes mellitus, the significance of difference in percentages suggests diabetes mellitus as a cause of differences in body composition. Disability in chewing was identified as a significantly associated factor for high fat mass among older women. Further research on oral health is required to understand the impact on food choices and body composition.

As consistent findings from other parts of the world, in Northwest Ethiopia [35] and France [28] age ≥ 70 years was significantly associated with undernutrition. The current data are in line with existing literature, indicating that tooth loss and not wearing dentures are associated factors of undernutrition [36].

Key factors having a significant inverse association with low muscle mass in men, included, age ≥ 70 years, no school education or up to grade 5, presence of diabetes, disability in vision, little or no responsibility in food shopping and not getting nutritional advice from media. As obesity is a behavioral risk factor of getting non-communicable diseases, such as diabetes, people with diabetes are generally obese and they tend to develop a high fat mass as well as a high muscle mass. As we defined low muscle mass as the bottom 25%, they (i.e., those with DM) may be less likely to be associated with low muscle mass in this sample. Low education is also associated with unskilled occupations, such as manual labor, and this may also explain the reduced likelihood of association with low muscle mass. No education or up to grade 5, disability in chewing and little or no responsibility in food shopping were significantly associated with reduced odds of high fat mass in men. This difference in gender in contrast to women, is interesting and is not completely understood. Plausible mechanisms exist. For example, low educational level and not connected to media may be associated with manual occupation and thus likely to have reduced risk of both low muscle mass and high fat mass in older age. Our study is not powered to test this hypothesis, however, how socioeconomic factors influence later life nutrition should be further explored. Future research is needed to explore the relationship between body composition and other factors, such as the level of physical activity of older people.

### 4.3. Strengths and Limitations

Our study has several strengths. Multistage cluster sampling techniques with probability proportionate to the size, are used in community-based surveys enabling the sample to be representative. The questionnaire was assessed by a panel of experts in the field of nutrition and elderly care, and it was interviewer administered to clarify the questions from older people. Most questions were structured to minimize interviewer bias. Composite criterion to identify undernutrition could explore the hidden burden among older people.

We also acknowledge several limitations. Unlike cohort studies, the cross-sectional nature of our study limited the exploration of the temporal relationship between associated factors and undernutrition. Exclusion of some older people with chronic diseases, such as cancers, neurodegenerative disorders and chronic renal failure, could have underestimated the results of the study as these chronic diseases are associated with undernutrition. The BIA measurements and the cut-off levels used in our study were based on European population data and therefore the prevalence of undernutrition may be higher than observed in this study if the Asian Working Group for Sarcopenia was applied [37]. Due to the relatively smaller number of men in the study, some associated factors among men were under explored and non-significant results may be prone to type II error. We have avoided selection algorithms, as we are interested in all the variables presented in the tables, rather than seeking a most parsimonious prediction model. Including correlated variables in a regression risks true effects appearing as non-significant as effects are estimated after adjustment for other variables. However, we do not make claims about effects significant in a univariate regression becoming non-significant in the multivariate model. Backward selection can lead to other problems we wished to avoid [38,39]. We did not formally assess the multicollinearity and auto-correlation and this may have attenuated results due to an element of over adjustment. Our study has used anthropometric and body composition to assess undernutrition. We were unable to incorporate all the anthropometric measurements, such as waist circumference, into our study to lessen the participant burden. Comprehensive assessment of nutritional status could ideally also include clinical and biochemical assessments and this could be a focus for future work.

## 5. Conclusions

Prevalence of undernutrition among older people is high in Sri Lanka. We identified determinants of undernutrition and factors that are associated with low muscle mass and high fat mass among community dwelling older men and women in a low-and middle-income country setting, providing useful information for public health interventions in targeting these potential risk factors.

## Figures and Tables

**Figure 1 geriatrics-07-00026-f001:**
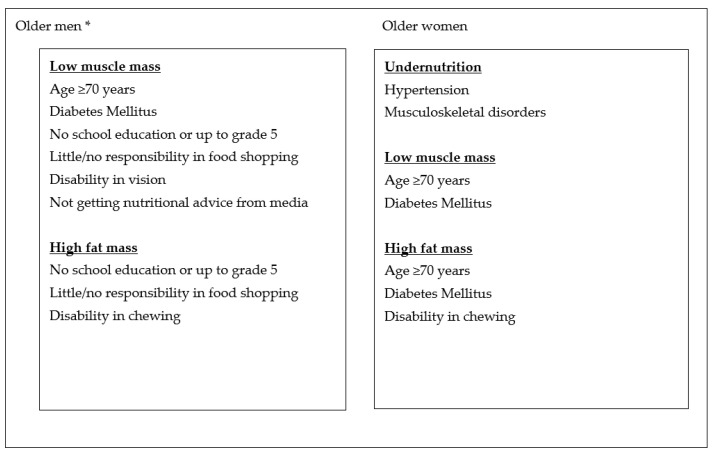
Summary of the factors associated with undernutrition, low muscle and high fat mass among older men and women. * Associated factors have a significant inverse association.

**Table 1 geriatrics-07-00026-t001:** Multiple comparison of factors among older men and women in the sample.

	Women 555/800		Men 245/800		Chi-Square Value (df = 1)	Effect Size (Phi Value)	*p* Value
Factor	N	%	N	%			
Age—equal to or above 70 years	184	33.2	101	41.2	4.828	0.08	0.028
Ethnicity—Sinhalese	547	98.6	230	93.9	13.338	0.13	0.001
Marital status—widowed, divorced or unmarried	204	36.8	7	2.9	1.006	0.04	0.001
Living environment—urban	212	38.5	53	21.6	21.055	0.16	0.001
Level of school education—none or up to grade 5	87	15.7	32	13.1	0.918	0.03	0.338
Unemployment	498	89.7	177	72.2	39.415	0.22	0.001
Not having a monthly income	390	70.3	101	41.2	60.489	0.27	0.001
Presence of diabetes	205	36.9	88	35.9	0.076	0.01	0.783
Presence of hypertension	242	43.6	92	37.6	2.560	0.06	0.110
Presence of heart disease	56	10.1	33	13.5	1.963	0.05	0.161
Presence of asthma/COPD	30	5.4	7	2.9	2.502	0.06	0.114
Disability in hearing	87	15.7	34	13.9	0.428	0.02	0.513
Disability in vision	437	78.7	178	72.7	3.541	0.07	0.060
Disability in chewing	137	22.4	42	17.1	5.566	0.08	0.018
Presence of musculoskeletal disorders	124	22.3	43	17.6	2.362	0.05	0.124
Current betel chewing	37	6.7	72	29.4	74.559	0.31	0.001
No responsibility in food shopping	415	74.8	208	84.9	16.109	0.14	0.001
No responsibility in planning meals	442	79.4	190	77.6	0.447	0.02	0.504
No responsibility in preparing meals	495	89.2	59	24.1	3.389	0.07	0.001
Skipping meals	114	20.5	50	20.4	0.002	0.00	0.966
Getting nutritional advice from GP	483	87.0	223	91.0	2.614	0.06	0.106
Getting nutritional advice from hospital	371	66.2	191	80.0	14.243	0.13	0.001
Getting nutritional advice from media	502	90.5	229	93.5	1.966	0.05	0.161

**Table 2 geriatrics-07-00026-t002:** Univariate and multivariate regression analysis of associated factors, odds ratios and their likelihood for undernutrition among older women.

	Women 262/555				
	Univariate Model			Multivariate Model	
Factor	OR (95% CI)	Effect Size (Cohen’s d Value)	*p* Value	OR (95% CI)	*p* Value
Age equal to or more than 70 years	1.56 (1.07–2.19)	0.114	0.018	0.61 (0.39–0.94)	0.027
Ethnicity—Sinhalese	2.72 (0.54–13.58)	0.239	0.20	0.42 (0.07–2.30)	0.32
Marital status—widowed, divorced or unmarried	1.02 (0.72–1.44)	0.005	0.90	1.01 (0.69–1.49)	0.96
Urban living environment	1.39 (0.99–1.97)	0.079	0.06	0.86 (0.57–1.30)	0.48
No school education or up to grade 5	0.89 (0.56–1.41)	0.028	0.63	1.47 (0.87–2.47)	0.15
Unemployment	1.26 (0.72–2.19)	0.055	0.41	0.92 (0.41–1.84)	0.81
Not having a monthly income	1.32 (0.91–1.90)	0.066	0.14	0.81 (0.50–1.30)	0.38
Presence of diabetes	1.04 (0.74–1.47)	0.009	0.83	0.85 (0.59–1.28)	0.43
Presence of hypertension	0.56 (0.40–0.79)	0.138	0.001	1.97 (1.36–2.88)	0.001
Presence of heart disease	0.89 (0.51–1.55)	0.038	0.68	1.23 (0.67–2.26)	0.50
Presence of asthma/COPD	1.13 (0.54–2.35)	0.029	0.75	1.14 (0.51–2.51)	0.75
Presence of disability in hearing	0.84 (0.53–1.34)	0.042	0.47	1.37 (0.78–2.40)	0.27
Presence of disability in vision	0.91 (0.60–1.36)	0.023	0.63	1.25 (0.80–1.95)	0.32
Presence of disability in chewing	1.55 (1.05–2.29)	0.105	0.026	0.56 (0.34–0.90)	0.018
Presence of musculoskeletal disorders	0.54 (0.35–0.81)	0.137	0.003	2.19 (1.36–3.53)	0.001
Current betel chewing	1.06 (0.55–2.07)	0.014	0.86	1.05 (0.51–2.15)	0.89
Little or no responsibility in food shopping	1.71 (1.16–2.51)	0.128	0.006	0.56 (0.27–1.18)	0.13
Little or no responsibility in planning meals	1.61 (1.06–2.44)	0.114	0.024	1.36 (0.67–3.24)	0.49
Little or no responsibility in preparing meals	1.65 (0.96–2.84)	0.115	0.07	0.78 (0.38–1.56)	0.48
Skipping meals	0.96 (0.64–1.46)	0.009	0.86	0.87 (0.55–1.37)	0.55
Not getting nutritional advice from GP	1.48 (0.89–2.40)	0.094	0.13	0.78 (0.43–1.39)	0.39
Not getting nutritional advice from hospital	0.79 (0.56–1.12)	0.056	0.19	1.31 (0.85–2.02)	0.22
Not getting nutritional advice from media	1.09 (0.62–1.92)	0.020	0.77	0.99 (0.52–1.89)	0.98

Regression analysis was unable to be conducted for older men due to smaller frequencies. Hosmer–Lemeshow goodness of fit value for regression models was 5.502 (*p* = 0.70).

**Table 3 geriatrics-07-00026-t003:** Univariate and multivariate regression analysis of associated factors, odds ratios and their likelihood for low muscle mass among older men and women.

	Women 143/555					Men 63/245				
	Univariate Model			Multivariate Model		Univariate Model			Multivariate Model	
Factor	OR (95% CI)	Effect Size (Cohen’s d Value)	*p* Value	OR (95% CI)	*p* Value	OR (95% CI)	Effect Size (Cohen’s d Value)	*p* Value	OR (95% CI)	*p* Value
Age equal to or more than 70 years	0.66 (0.42–1.0)	0.099	0.05	1.79 (1.18–3.34)	0.009	1.84 (1.03–3.29)	0.146	0.037	0.43 (0.20–9.1)	0.028
Ethnicity—Sinhalese	0.57 (0.13–2.4)	0.134	0.44	0.44 (0.09–2.04)	0.29	0.36 (0.13–1.06)	0.244	0.055	2.19 (0.64–7.52)	0.21
Marital status—widowed, divorced or unmarried	0.79 (0.53–1.18)	0.056	0.26	1.12 (0.75–1.76)	0.61	2.22 (0.48–10.22)	0.190	0.29	0.34 (0.05–2.42)	0.28
Urban living environment	1.19 (0.81–1.8)	0.042	0.38	0.85 (0.53–1.36)	0.49	0.81 (0.39–1.66)	0.050	0.56	0.93 (0.35–2.47)	0.88
No school education or up to grade 5	1.04 (0.62–1.8)	0.009	0.88	0.64 (0.35–1.15)	0.13	2.6 (1.20–5.60)	0.228	0.012	0.27 (0.09–0.76)	0.014
Unemployment	0.61 (0.34–1.09)	0.118	0.09	1.28 (0.60–2.70)	0.52	1.47 (0.75–2.89)	0.092	0.25	1.89 (0.65–5.47)	0.24
Not having a monthly income	0.69 (0.46–1.03)	0.089	0.07	1.17 (0.69–1.90)	0.56	2.01 (1.12–3.59)	0.167	0.017	0.67 (0.28–1.63)	0.38
Presence of diabetes	0.63 (0.42–0.96)	0.110	0.03	1.77 (1.10–2.84)	0.017	2.12 (1.18–3.81)	0.179	0.011	0.34 (0.15–0.77)	0.010
Presence of hypertension	1.56 (1.06–2.28)	0.106	0.023	0.61 (0.39–0.92)	0.020	1.35 (0.75–2.42)	0.072	0.31	0.93 (0.42–2.05)	0.86
Presence of heart disease	1.17 (0.63–2.20)	0.038	0.61	0.88 (0.46–1.69)	0.69	1.09 (0.48–2.56)	0.021	0.83	0.99 (0.35–2.77)	0.99
Presence of asthma/COPD	1.25 (0.56–2.80)	0.053	0.59	0.60 (0.25–1.43)	0.25	1.16 (0.21–6.13)	0.035	0.86	3.22 (0.44–23.68)	0.25
Presence of disability in hearing	1.04 (0.62–1.76)	0.009	0.88	0.95 (0.51–1.78)	0.88	1.71 (0.79–3.69)	0.128	0.17	0.63 (0.21–1.88)	0.41
Presence of disability in vision	0.91 (0.58–1.45)	0.023	0.70	0.86 (0.52–1.78)	0.57	3.93 (1.69–9.15)	0.327	0.001	0.24 (0.08–0.69)	0.008
Presence of disability in chewing	0.76 (0.48–1.20)	0.066	0.23	1.69 (0.96–2.98)	0.07	1.79 (0.88–3.64)	0.139	0.10	0.75 (0.25–2.20)	0.60
Presence of musculoskeletal disorders	2.03 (1.32–3.12)	0.169	0.001	0.39 (0.24–0.65)	0.001	1.31 (0.63–2.70)	0.064	0.45	1.85 (0.59–5.7)	0.28
Current betel chewing	0.78 (0.35–1.75)	0.059	0.55	1.00 (0.42–2.38)	0.99	0.94 (0.50–1.78)	0.015	0.87	0.94 (0.40–2.19)	0.88
Little or no responsibility in food shopping	0.75 (0.49–1.15)	0.059	0.18	0.75 (0.34–1.67)	0.48	0.33 (0.16–0.69)	0.265	0.002	0.11 (0.01–0.67)	0.017
Little or no responsibility in planning meals	0.76 (0.48–1.20)	0.066	0.23	0.92 (0.36–2.37)	0.87	0.51 (0.22–0.98)	0.134	0.040	1.87 (0.37–9.48)	0.45
Little or no responsibility in preparing meals	0.66 (0.37–1.17)	0.099	0.15	0.82 (0.38–1.76)	0.61	1.10 (0.57–2.13)	0.023	0.77	1.65 (0.68–3.99)	0.26
Skipping meals	1.16 (0.73–1.80)	0.035	0.52	0.94 (0.56–1.58)	0.82	1.16 (0.58–2.33)	0.035	0.67	0.65 (0.26–1.62)	0.36
Not getting nutritional advice from GP	0.61 (0.36–1.03)	0.118	0.06	1.31 (0.71–2.45)	0.39	0.46 (0.18–1.14)	0.185	0.08	1.94 (0.57–6.6)	0.29
Not getting nutritional advice from hospital	1.11 (0.74–1.67)	0.025	0.61	0.92 (0.56–1.51)	0.75	1.08 (0.52–2.24)	0.018	0.82	0.58 (0.22–1.54)	0.27
Not getting nutritional advice from media	1.07 (0.56–2.07)	0.016	0.83	0.72 (0.35–1.51)	0.39	5.57 (0.72–43.04)	0.410	0.06	0.07 (0.01–0.95)	0.045

Hosmer–Lemeshow goodness of fit values for regression models among men and women were 7.503 (*p* = 0.48) and 13.13 (*p* = 0.11).

**Table 4 geriatrics-07-00026-t004:** Univariate and multivariate regression analysis of associated factors, odds ratios and their likelihood for high fat mass among older men and women.

	Women 139/555					Men 66/245				
	Univariate Model			Multivariate Model		Univariate Model			Multivariate Model	
Factor	OR (95% CI)	Effect Size (Cohen’s d Value)	*p* Value	OR (95% CI)	*p* Value	OR (95% CI)	Cohen’s d Value	*p* Value	OR (95% CI)	*p* Value
Age equal to or more than 70 years	0.66 (0.43–1.05)	0.099	0.06	2.05 (1.21–3.47)	**0.007**	0.90 (0.50–1.60)	0.025	0.72	1.31 (0.62–2.70)	0.48
Ethnicity—Sinhalese	1.00 (0.20–5.02)	0.000	0.99	0.75 (0.13–4.21)	0.74	1.50 (0.41–5.52)	0.097	0.53	0.77 (0.17–3.40)	0.73
Marital status—widowed, divorced or unmarried	1.12 (0.76–1.67)	0.027	0.55	0.66 (0.42–1.05)	0.08	1.08 (0.20–5.70)	0.018	0.93	1.90 (0.20–18.12)	0.56
Urban living environment	1.48 (1.00–2.19)	0.094	0.046	0.72 (0.45–1.16)	0.18	1.09 (0.55–2.15)	0.021	0.80	0.76 (0.32–1.82)	0.54
No school education or up to grade 5	1.17 (0.69–1.96)	0.038	0.55	0.50 (0.26–0.92)	0.026	2.41 (1.17–5.17)	0.210	0.022	0.29 (0.10–0.85)	0.024
Unemployment	0.76 (0.42–1.39)	0.066	0.38	1.12 (0.52–2.44)	0.77	0.76 (0.41–1.40)	0.066	0.39	2.07 (0.82–5.20)	0.12
Not having a monthly income	0.74 (0.49–1.12)	0.079	0.15	1.08 (0.63–1.85)	0.78	1.07 (0.60–1.89)	0.016	0.82	0.89 (0.38–2.13)	0.81
Presence of diabetes	0.57 (0.37–0.86)	0.134	0.007	2.20 (1.35–3.59)	0.002	1.46 (0.82–2.60)	0.090	0.19	0.69 (0.32–1.49)	0.36
Presence of hypertension	1.96 (1.33–2.89)	0.161	0.001	0.44 (0.28–0.68)	0.44	1.44 (0.81–2.56)	0.087	0.21	1.25 (0.59–2.60)	0.56
Presence of heart disease	1.48 (0.81–2.69)	0.094	0.19	0.72 (0.38–1.39)	0.33	1.42 (0.65–3.13)	0.084	0.37	0.52 (0.20–1.33)	0.17
Presence of asthma/COPD	1.53 (0.70–3.36)	0.102	0.28	0.46 (0.19–1.09)	0.08	1.08 (0.20–5.74)	0.018	0.92	2.58 (0.38–17.36)	0.33
Presence of disability in hearing	0.94 (0.55–1.61)	0.107	0.83	1.02 (0.54–1.93)	0.95	1.35 (0.62–2.96)	0.072	0.44	1.24 (0.41–3.74)	0.70
Presence of disability in vision	0.92 (0.58–1.46)	0.020	0.73	0.80 (0.48–1.33)	0.80	1.34 (0.65–2.38)	0.071	0.51	0.66 (0.28–1.50)	0.66
Presence of disability in chewing	0.56 (0.34–0.91)	0.138	0.019	2.39 (1.30–4.40)	0.005	2.13 (1.05–4.28)	0.181	0.030	0.34 (0.13–0.94)	0.037
Presence of musculoskeletal disorders	1.37 (0.88–2.14)	0.075	0.16	0.59 (0.35–0.98)	0.045	1.21 (059–2.50)	0.045	0.59	1.11 (0.38–3.18)	0.85
Current betel chewing	0.68 (0.29–1.59)	0.092	0.37	1.29 (0.52–3.2)	0.58	0.70(0.37–1.34)	0.085	0.28	1.65 (0.73–3.76)	0.23
Little or no responsibility in food shopping	0.86 (0.56–1.33)	0.036	0.51	1.20 (0.50–2.89)	0.67	0.36 (0.18–0.75)	0.244	0.005	0.05 (0.01–0.35)	0.002
Little or no responsibility in planning meals	0.76 (0.48–1.21)	0.066	0.25	0.58 (0.21–1.57)	0.28	0.62 (0.33–1.19)	0.114	0.15	4.43 (0.79–24.73)	0.09
Little or no responsibility in preparing meals	0.76 (0.42–1.36)	0.066	0.35	0.97 (0.44–2.14)	0.94	1.01 (0.52–1.96)	0.002	0.97	1.57 (0.69–3.58)	0.29
Skipping meals	1.36 (0.86–2.15)	0.074	0.19	0.69 (0.41–1.16)	0.16	0.72 (0.34–1.50)	0.078	0.38	1.15 (0.46–2.86)	0.77
Not getting nutritional advice from GP	0.62 (0.37–1.07)	0.114	0.08	1.47 (0.78–2.77)	0.23	1.28 (0.45–3.67)	0.059	0.64	1.24 (0.34–4.54)	0.74
Not getting nutritional advice from hospital	1.00 (0.67–1.51)	0.000	0.97	1.03 (0.62–1.69)	0.91	0.71 (0.36–1.40)	0.082	0.31	1.23 (0.51–2.98)	0.65
Not getting nutritional advice from media	0.83 (0.44–1.56)	0.044	0.56	0.96 (0.46–1.97)	0.90	0.79 (0.27–2.39)	0.056	0.69	2.03 (0.47–8.70)	0.34

Hosmer–Lemeshow goodness of fit values for regression models among men and women were 6.452 (*p* = 0.59) and 12.91 (*p* = 0.12).

## Data Availability

The data presented in this study are available on request from the corresponding author. The data are not publicly available due to the privacy of participants and ethical reasons.

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
