# Peer review of "Determinants of Undernutrition and Associated Factors of Low Muscle Mass and High Fat Mass among Older Men and Women in the Colombo District of Sri Lanka"

_geriatrics, 2022, doi:10.3390/geriatrics7020026_

Round 1

Reviewer 1 Report

Thank you for completing the amendments.

Author Response

Thank you very much for your valuable comments given to us to improve our article.

Reviewer 2 Report

All my comments have been addressed. I have just one comment.

Page 17 lines 83-84: The authors state that "BIA measurements and the cut-off levels were based on the European population data due to non-availability of validated cut-offs for the Asians at the time of the study". I have some doubts about this statement since the cut-off points to define sarcopenia in Asian people are available since 2014 (Chen et al. 2014 doi: 10.1016/j.jamda.2013.11.025) with an update released in 2019 by the Asian Working Group for Sarcopenia (Chen et al. 2019 doi: 10.1016/j.jamda.2019.12.012). 

Author Response

This manuscript is a resubmission of an earlier submission. The following is a list of the peer review reports and author responses from that submission.

Round 1

Reviewer 1 Report

Dear Authors,

Thank you for submitting your manuscript for publication in the journal Geriatrics and providing me with the opportunity to review it. It was a pleasure to review your work. Please complete the following amendments/clarifications before resubmitting your manuscript:

  1. The abstract needs to focus on what was done, found and concluded. In line with that, the objectives might need to be reconsidered as a simple clear statement of aim, the short form of aOR needs to be introduced, and the conclusion can be further informative, in view of overall and collective findings.
  2. Within the introduction, undernutrition is defined, but the criteria for diagnosis of undernutrition is not discussed. I strongly recommend producing a table with different diagnostic definitions of undernutrition in this population with reference and perhaps an appraisal and justification of why the current criteria were used within the current study. The narrative in lines 44-49 needs further expansion and clarification.
  3. Line 40, there is an explanation of sarcopenia with no reference, and it does not clearly contribute to the point discussed. Please clarify and improve the narrative.
  4. Prevalence of malnutrition in older adults in clinical, care and community-dwelling settings are reported, if those studies were conducted, why there was a need for the current study. Please clarify within the manuscript. To elucidate the task, the previous literature needs to be critically appraised and the rationale for the conduct of the current investigation needs clarification. For this, the author might want to consider producing a table about the previous studies, author, date, setting, key findings, strengths and limitations to support the rationale that will be added.
  5. The term 'sarcopenic obesity has been mentioned. The term, its definition and the burden of sarcopenia and obesity amongst this population need to be clarified.
  6. Please clarify the conceptual association between undernutrition and sarcopenic obesity within a new paragraph within the introduction.
  7. The aims are mentioned unclearly and scattered within two paragraphs. Please clarify the aim (or primary and secondary aims) in a clear and concise, focused and scientific statement.
  8. If the study was conducted only in one district (i.e. Colombo District) in Sri Lanka, this needs to be amended in the title and across the manuscript as the findings do not necessarily represent the rest of Sri Lanka.
  9. Lines 62-65 or 69-73, please clarify the statements related to the recruitment process including who were the participants invited (inclusion and exclusion criteria), how they were identified and recruited, etc in view of the representation of the entire target population within the district.
  10. Please clarify who are the assessors, how they were trained and how the target level of 5% precision (or additional inter and intra - examiner variation) targets were achieved.
  11. Please provide additional statements about the validity of using BIA and in particular Omron scales (HBF – 212, Japan) for assessment of body composition in older adults.
  12. Hydration substantially affects the assessment of BIA and this is especially an important consideration in the assessment of body composition in Geriatrics. How this was taken into consideration?
  13. Please provide a clear explanation about why dietary intake was not assessed in the study of undernutrition amongst older adults.
  14. lines 100-105, please clarify how normal distribution was assessed, what variables were not normally distributed and the procedure for analysis of those varaiables.
  15. What are the diagnostic criteria for the assessment of clinical conditions such as diabetes amongst this older adults population?
  16. In an older population with more than 90% diabetes, the body composition, in particular, gets affected by the disease. Please clarify the role of the disease assessed on the key findings of the study.
  17. Was waist circumference assessed?
  18. Please provide a figure or a table to discuss the key findings of the current study in comparison with the previous studies.
  19. Please provide a schematic or figure to showcase the factors affecting undernutrition in this population as found within the current study.
  20. Lines 52-55, the limitations can be discussed more clearly, more thoroughly and more reflectively and the paragraph is needed to be expanded with several points highlighted within the lines above.

Reviewer 2 Report

In the present article, the authors explored the prevalence, determinants and associated factors of undernutrition in a population of older adults living in Sri Lanka. Overall, it is a nice research idea. However, I think multiple points need to be addressed. English editing is needed.

Page 1; line 20: How were defined “musculoskeletal disabilities”?

Page 2; line 40: Please add the definition of sarcopenia (i.e. low muscle mass and strength leading to poor physical function).

Page 2; line 40-41: In fact, undernutrition can result in a loss of body weight and nutritional deficiencies leading to fatigue through lack of energy. To date, if the intake of energy and proteins is not adequate to meet individual needs, body fat and muscle are catabolized to provide energy with consequent symptoms like as fatigue or lack of energy (doi.org/10.1111/jgs.17393). It can be mentioned.

Page 2; line 42-43: Please rephrase.

Page 2; line 50-52: Additionally, a major variability in body fat distribution, in particular for what concerns ectopic and visceral fat, may be determined by ethnic differences. Obesity, defined according to BMI, may be underestimated in older adults, who may present an excess of adiposity within a normal/overweight body size (doi.org/10.3389/fendo.2020.581356).

Page 3; line 77: Please change “the status of undernutrition” to “nutritional status”

Page 3; line 78: Please change “mass” to “weight”

Page 3; line 86: Please change “BIA” to “body composition”

Page 3; line 91-92: From what I gather, the instrument used for this study provides only the percentage of body fat and muscle mass, while does not provide data about resistance, reactance and impedance, is that correct?

Page 3; line 95: Recently, the Global Leadership Initiative on Malnutrition (GLIM) pointed out that further research is needed regarding reference BMI data for the Asian population in clinical settings. However, they suggested a cut-off point for low BMI <20 Kg/m2 for those Asian people aged 70+ while the BMI cut-off point <18.5 Kg/m2 for those people aged <70 years may be fine.

Page 3; line 96-97: “whilst for muscle mass was defined using values obtained by Tichet et al.[19]”. These values are referred to the French population and may not apply to Asian people. To date, the Asian Working Group for Sarcopenia (AWGS) released a consensus with specific cut-points for defining low muscle mass and strength in the Asian population (DOI: 10.1016/j.jamda.2019.12.012).

Page 14; line 15-16: In addition to anthropometry and recent weight loss, the MNA includes also other components (i.e., reduced mobility, decrease in food intake, acute disease or psychological stress over the past 3 months).

Page 15; line 39: For this reason, the AWGS proposed the criteria for low muscle mass based on Asian data.

Round 2

Reviewer 2 Report

All  my comments have been addressed